# Submodular Attribute Selection for Action Recognition in Video

**Jinging Zheng**
UMIACS, University of Maryland
College Park, MD, USA
zjngjng@umiacs.umd.edu

**Zhuolin Jiang**
Noah's Ark Lab
Huawei Technologies
zhuolin.jiang@huawei.com

**Rama Chellappa**
UMIACS, University of Maryland
College Park, MD, USA
rama@umiacs.umd.edu

**P. Jonathon Phillips**
National Institute of Standards and Technology
Gaithersburg, MD, USA
jonathon.phillips@nist.gov

## Abstract

In real-world action recognition problems, low-level features cannot adequately characterize the rich spatial-temporal structures in action videos. In this work, we encode actions based on attributes that describes actions as high-level concepts *e.g.*, jump forward or motion in the air. We base our analysis on two types of action attributes. One type of action attributes is generated by humans. The second type is data-driven attributes, which are learned from data using dictionary learning methods. Attribute-based representation may exhibit high variance due to noisy and redundant attributes. We propose a discriminative and compact attribute-based representation by selecting a subset of discriminative attributes from a large attribute set. Three attribute selection criteria are proposed and formulated as a submodular optimization problem. A greedy optimization algorithm is presented and guaranteed to be at least (1-1/e)-approximation to the optimum. Experimental results on the Olympic Sports and UCF101 datasets demonstrate that the proposed attribute-based representation can significantly boost the performance of action recognition algorithms and outperform most recently proposed recognition approaches.

## 1 Introduction

Action recognition in real-world videos has many potential applications in multimedia retrieval, video surveillance and human computer interaction. In order to accurately recognize human actions from videos, most existing approaches developed various discriminative low-level features, including spatio-temporal interest point (STIP) based features [8, 15], shape and optical flow-based features [19, 5], and trajectory-based representations [28, 33]. Because of large variations in viewpoints, complicated backgrounds, and people performing the actions differently, videos of an action vary greatly. A result of this variability is that conventional low-level features are not able to characterize the rich spatio-temporal structures in real-world action videos. Inspired by recent progress on object recognition [6, 14], multiple high-level semantic concepts called action attributes were introduced in [20, 17] to describe the spatio-temporal evolution of the action, object shapes and human poses, and contextual scenes. Since these action attributes are relatively robust to changes in viewpoints and scenes, they bridge the gap between low-level features and class labels. In this work, we focus on improving action recognition performance of attribute-based representations.

Even though attribute-based representation appear effective for action recognition, they require humans to generate a list of attributes that may adequately describe a set of actions. From this list, humans then need to assign the action attributes to each class. Previous approaches [20, 17] simply used all the given attributes and ignored the difference in discriminative capability among attributes. This caused two major problems. First, a set of human-labeled attributes may be not be able to

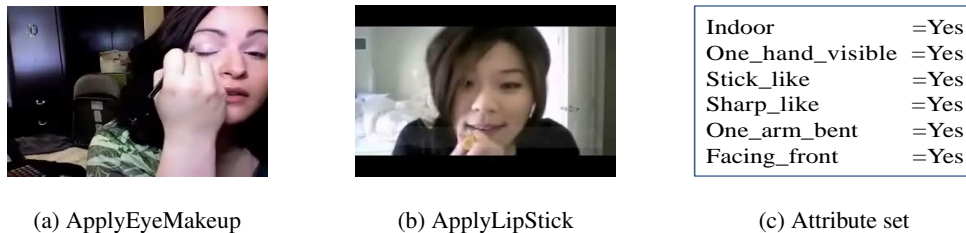

| Indoor | =Yes |
| One_hand_visible | =Yes |
| Stick_like | =Yes |
| Sharp_like | =Yes |
| One_arm_bent | =Yes |
| Facing_front | =Yes |

(a) ApplyEyeMakeup  (b) ApplyLipStick  (c) Attribute set

Figure 1: **Key frames from two actions "ApplyEyeMakeup" and "ApplyLipStick" and the associated attribute set that the two actions share.**

represent and distinguish a set of action classes. This is because humans subjectively annotate action videos with arbitrary attributes. For example, consider the two classes "ApplyEyeMakeup" and "ApplyLipStick" in UCF101 action dataset [30] shown in Figure 1. They have the same human-labeled attribute set and cannot be distinguished from one another. Second, some manually labeled attributes may be noisy or redundant which leads to degradation in action recognition performance. In addition, their inclusion also increases the feature extraction time. Thus, it would be beneficial to use a smaller subset of attributes while achieving comparable or even improved performance.

To overcome the first problem, we propose another type of attributes that we call *data-driven* attributes. We show that data data-driven attributes are complementary to human-labeled attributes. Instead of using clustering-based algorithms to discover data-driven attributes as in [20], we propose a dictionary-based sparse representation method to discover a large data-driven attribute set. Our learned attributes are more suited to represent all the input data points because our method avoids the problem of hard assignment of data points to clusters. To address the attribute selection problem, we propose to select a compact and discriminative set of attributes from a large set of attributes. Three attribute selection criteria are proposed and then combined to form a submodular objective function. Our method encourages the selected attributes to have strong and similar discrimination capability for all pairs of actions. Furthermore, our method maximizes the sum of maximum coverage that each pairwise class can obtain from the selected attributes.

## 2   Related Work

**Attribute-based representation for action recognition**: Recently, several attribute-based representations have been proposed for improving action recognition performance. Liu et al. [20] modeled attributes as latent variables and searched for the best configuration of attributes for each action using latent SVMs. However, the performance may drop drastically when some attributes are too noisy or redundant. This is because pretrained attribute classifiers from these noisy attributes perform poorly. Li et al. [17] decomposed a video sequence into short-term segments and characterized segments by the dynamics of their attributes. However, since attributes are defined over the entire action video instead of short-term segments, different decomposition of video segments may obtain different attribute dynamics.

Another line of work similar to attribute-based methods is based on learning different types of mid-level representations. These mid-level representations usually identify the occurrence of semantic concepts of interest, such as scene types, actions and objects. Fathi et al. [7] proposed to construct mid-level motion features from low-level optical flow features using AdaBoost. Wang et al. [35] modeled a human action as a global root template and a constellation of several parts. Raptis et al. [27] used trajectory clusters as candidates for the parts of an action and assembled these clusters into an action class by graphical modeling. Jain et al. [10] presented a new mid-level representation for videos based on discriminative spatio-temporal patches, which are automatically mined from videos using an exemplar-based clustering approach.

**Submodularity**: Submodular functions are a class of set functions that have the the property of *diminishing returns* [24]. Given a set $E$, a set function $F : 2^E \rightarrow R$ is submodular if $F(A \cup v) - F(A) \geq f(B \cup v) - F(B)$ holds for all $A \subseteq B \subseteq E$ and $v \in E \setminus B$. The diminishing returns mean that the marginal value of the element $v$ decreases if used in a later stage. Recently, submodular functions have been widely exploited in various applications, such as sensor placements [13], superpixel segmentation [22], document summarization [18], and feature selection [3, 23]. Liu et al. [23] presented a submodular feature selection method for acoustic score spaces based on existing facility location and saturated coverage functions. Krause et al. [12] de-

veloped a submodular method for selecting dictionary columns from multiple candidates for sparse representation. Iyer et al. [9] designed a new framework for both unconstrained and constrained submodular function optimization. Streeter et al. [31] proposed an online algorithm for maximizing submodular functions. Different from these approaches, we define a novel submodular objective function for attribute selection. Although we only evaluate our approach for action recognition, it can be applied to other recognition tasks that use attribute descriptions.

## 3  Submodular Attribute Selection

In this section, we first propose three attribute selection criteria. In order to satisfy these criteria, we define a submodular function based on entropy rate of a random walk and a weighted maximum coverage function. Then we introduce algorithms for the detection of human-labeled attributes and extraction of data-driven attributes.

### 3.1  Attribute Selection Criteria

Assume that we have $C$ classes and a large attribute set $\mathcal{P} = \{a_1, a_2, .., a_M\}$ which contains $M$ attributes. The set that includes all combinations of pairwise classes is represented by $\mathcal{U} = \{u_1(1,1), u_2(1,2), ..., u_l(i,j), ..., u_L(C-1,C)\}$ where $u_l(i,j), i < j$ denotes the pairwise combination of classes $i$ and $j$, $l$ is the index of this combination in $\mathcal{U}$, and $L = C \times (C-1)/2$ is the total number of all possible pairwise classes. Here we propose to use the Fisher score to construct an attribute contribution matrix $A \in R^{M \times L}$, where an entry $A_{d,l}$ represents the discrimination capability of attribute $a_d$ for differentiating the class pair $(i,j)$ indexed by $u_l(i,j)$. Specifically, given the attribute $a_d$ and class pair $(i,j)$, let $\mu_k^d$ and $\sigma_k^d$ be the mean and standard deviation of $k$-th class and $\mu^d$ be the mean of samples from both classes $i$ and $j$ corresponding to $d$-th attribute. The Fisher score of attribute $a_d$ for differentiating the class pair $(i,j)$ is computed as follows: $A_{d,l(i,j)} = \frac{\sum_{k=i,j} n_k (\mu_k^d - \mu^d)^2}{\sum_{k=i,j} n_k \sigma_k^2}$ where $l$ is the index of pairwise classes $(i,j)$ in $\mathcal{U}$, and $n_k$ is the number of points from class $k$. Note that different methods can be used to measure the discrimination capability of $a_d$, such as mutual information and T-test.

Given $A$, we can obtain a row vector $r$ by summing up its elements from each column that are in rows corresponding to selected attributes $\mathcal{S}$. An example of vector $r$ is shown in Figure 2a. We would like to have $r$ satisfy two selection criteria: (1) each entry of $r$ should be as large as possible; and (2) the variance of all entries of $r$ should be small. The first criterion encourages $\mathcal{S}$ to provide as much discrimination capability as possible for each pairwise classes. The second criterion makes $\mathcal{S}$ have similar discrimination capability for each pairwise classes. These two criteria can be satisfied by maximizing the entropy rate of a random walk on the proposed graphs. Meanwhile, since some attributes may well differentiate the same collection of pairwise classes, it would be redundant to select all these attributes. In other words, one combination of pairwise classes may be repeatedly "covered" (differentiated) by multiple attributes. It is better to select other attributes which can differentiate "uncovered" combinations of pairwise classes. Therefore, we propose the third criterion: the sum of maximum discrimination capability that each pairwise classes can obtain from the selected attributes should be maximized. We will model it as a weighted maximum coverage problem and encourage $\mathcal{S}$ to have a maximum coverage of all pairwise classes.

### 3.2  Entropy Rate-based Attribute Selection

In order to achieve the first two criteria, we need to construct an undirected graph and maximize the entropy rate of a random walk on this graph. We aim to obtain a subset $\mathcal{S}$ so that the attribute-based representation has good discrimination power.

**Graph Construction**: We use $G = (V, E)$ to denote an undirected graph where $V$ is the vertex set, and $E$ is the edge set. The vertex $v_i$ represents class $i$ and the edge $e_{i,j}$ connecting class $i$ and $j$ represents that class $i$ and $j$ can be differentiated by the selected attribute subset $\mathcal{S}$ to some extent. The edge weight for $e_{i,j}$ is defined as $w_{i,j} = \sum_{d \in \mathcal{S}} A_{d,l}$, which represents the discrimination capability of $\mathcal{S}$ for differentiating class $i$ from class $j$. The edge weights are symmetric, i.e. $w_{i,j} = w_{j,i}$. In addition, we add a self-loop $e_{i,i}$ for each vertex $v_i$ of $G$. And the weight for self-loop $e_{i,i}$ is defined as $w_{i,i} = \sum_{d \in \mathcal{P} \setminus \mathcal{S}} A_{d,l}$. The total incident weight for each vertex is kept constant so that it produces a stationary distribution for the later proposed random walk on this graph. Note that the addition of these self-loops do not affect the selection of attributes and the graph will change with the selected subset $\mathcal{S}$. Figure 2 gives an example to illustrate the benefits of the entropy rate.

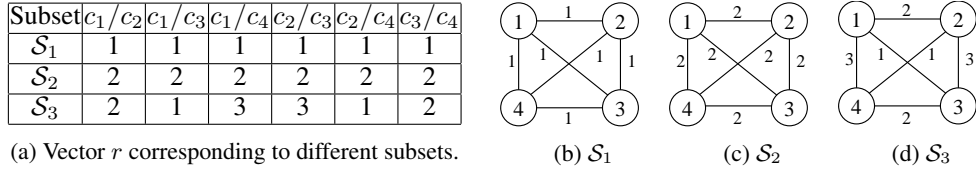

| Subset | $c_1/c_2$ | $c_1/c_3$ | $c_1/c_4$ | $c_2/c_3$ | $c_2/c_4$ | $c_3/c_4$ |
|--------|-----------|-----------|-----------|-----------|-----------|-----------|
| $\mathcal{S}_1$ | 1 | 1 | 1 | 1 | 1 | 1 |
| $\mathcal{S}_2$ | 2 | 2 | 2 | 2 | 2 | 2 |
| $\mathcal{S}_3$ | 2 | 1 | 3 | 3 | 1 | 2 |

(a) Vector $r$ corresponding to different subsets.          (b) $\mathcal{S}_1$          (c) $\mathcal{S}_2$          (d) $\mathcal{S}_3$

Figure 2: **The summations of different rows in the contribution matrix corresponding to three different selected subsets are provided in the left table and the corresponding undirected graphs are in the right figure.** We show the role of the entropy rate in selecting attributes which have large and similar discrimination capability for each pair of classes. The circles with numbers denote the corresponding class vertices and the numbers next to the edge denote the edge weights, which is a measure of the discrimination capability of selected attribute subset. The self-loops are not displayed. The entropy rate of the graph with large edge weights in (c) has a higher objective value than that of a graph with smaller edge weights in (b). The entropy rate of graph with equal edge weights in (c) has a higher objective value than that of the graph with different edge weights in (d).

**Entropy Rate**: Let $X = \{X_t | t \in T, X_t \in V\}$ be a random walk on the graph $G = (V, E)$ with nonnegative discrimination measure $w$. We use the random walk model from [2] with a transition probability defined as below:

$$p_{i,j}(\mathcal{S}) = \begin{cases} \frac{w_{i,j}}{w_i} = \frac{\sum_{d \in \mathcal{S}} A_{d,l}}{w_i} & \text{if } i \neq j \\ 1 - \frac{\sum_{k:k \neq i} w_{i,k}}{w_i} = \frac{\sum_{d \in \mathcal{P} \setminus \mathcal{S}} A_{d,l}}{w_i} & \text{if } i = j \end{cases} \tag{1}$$

where $\mathcal{S}$ is the selected attribute subset and $w_i = \sum_{m:e_{i,m} \in E} w_{i,m}$ is the sum of incident weights of the vertex $v_i$ including the self-loop. The stationary distribution for this random walk is given by $\mu = (\mu_1, \mu_2, ..., \mu_C)^T = (\frac{w_1}{w_0}, \frac{w_2}{w_0}, ..., \frac{w_C}{w_0})$ where $w_0 = \sum_{i=1}^{C} w_i$ is the sum of the total weights incident on all vertices. For a stationary 1st-order Markov chain, the entropy rate which measures the uncertainty of the stochastic process $X$ is given by: $\mathcal{H}(X) = \lim_{t \to \infty} H(X_t | X_{t-1}, X_{t-2}, ..., X_1) = \lim_{t \to \infty} H(X_t | X_{t-1}) = H(X_2 | X_1)$. More details can be found in [2]. Consequently, the entropy rate of the random walk $X$ on our proposed graph $G = (V, E)$ can be written as a set function:

$$\mathcal{H}(\mathcal{S}) = \sum_i u_i H(X_2 | X_1 = v_i) = -\sum_i u_i \sum_j p_{i,j}(\mathcal{S}) log(p_{i,j}(\mathcal{S})) \tag{2}$$

Intuitively, the maximization of the entropy rate will have two properties. First, it encourages the maximization of $p_{i,j}(\mathcal{S})$ where $i = 1, ..., C$ and $i \neq j$. This can make edge weights $w_{i,j}, i \neq j$ as large as possible, so class $i$ can be easily differentiated from other classes j (i.e., satisfying the first criteria). Second, it makes all class vertices have transition probabilities similar to other connected class vertices, so the discrimination capabilities of class $i$ from other classes are very similar (i.e., satisfying the second criteria). Maximizing the entropy rate of the random walk on the proposed graph can select a subset of attributes that are compact and discriminative for differentiating all pairwise classes.

**Proposition 3.1.** *The entropy rate of the random walk $\mathcal{H} : 2^M \to R$ is a submodular function under the proposed graph construction.*

The observation that adding an attribute in a later stage has a lower increase in the uncertainty establishes the submodularity of the entropy rate. This is because at a later stage, the increased edge weights from the added attribute will be shared with attributes which contribute to the differentiation of the same pair of classes. A detailed proof based on [22] is given in the supplementary section.

### 3.3 Weighted Maximum Coverage-based Attribute Selection

We consider a weighted maximum coverage function to achieve the last criteria that the selected subset $\mathcal{S}$ should maximize the coverage of all combinations of pairwise classes. For each attribute $a_d$, we define a coverage set $\mathcal{U}_d \subseteq \mathcal{U}$ which covers all the combinations of pairwise classes that attribute $a_d$ can differentiate. Meanwhile, for each element (combination) $u_l \in \mathcal{U}$ that is covered by $\mathcal{U}_d$, we define a coverage weight $w(\mathcal{U}_d, u_l) = A_{d,l}$. Given the universe set $\mathcal{U}$ and these coverage sets $\mathcal{U}_d, d = 1, ..., M$, the weighted maximum coverage problem is to select at most $K$ coverage sets, such that the sum of maximum coverage weight each element can obtain from $\mathcal{S}$ is maximized. The weighted maximum coverage function is defined as follows:

$$\mathcal{Q}(\mathcal{S}) = \sum_{u_l \in U} \max_{d \in \mathcal{S}} w(U_d, u_l) = \sum_{u_l \in U} \max_{d \in \mathcal{S}} A_{d,l}, \quad \text{s.t. } N_{\mathcal{S}} \leq K \tag{3}$$

| Attrs. | $c_1/c_2$ | $c_1/c_3$ | $c_1/c_4$ | $c_2/c_3$ | $c_2/c_4$ | $c_3/c_4$ |
|--------|-----------|-----------|-----------|-----------|-----------|-----------|
| $a_1$  | 2         | 2         | 0         | 1         | 1         | 0         |
| $a_2$  | 1         | 1         | 0         | 0         | 0         | 0         |
| $a_3$  | 0         | 0         | 1         | 0         | 0         | 2         |
| $a_4$  | 0         | 0         | 0         | 2         | 2         | 0         |

(a) Attribute contribution matrix $A$.

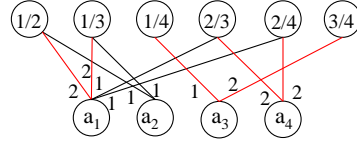

(b) Coverage graph

Figure 3: **An example of attribute contribution matrix is given in the left table and the corresponding coverage graph is in the right figure.** We show the role of weighted maximum coverage term in selecting attributes which have large coverage weights. Two numbers separated by a backslash in the top circles denote a pair of classes, while the bottom circles denote different attributes. The number next to one edge is the coverage weight associated with the class pair when covered by the corresponding attribute. The edge which provides maximum coverage weight for each class pair is in red color. We consider three attribute subsets $\mathcal{S}_1 = \{a_1, a_2\}, \mathcal{S}_2 = \{a_1, a_3\}, \mathcal{S}_3 = \{a_1, a_4\}$. $\mathcal{S}_2$ has a higher objective value than $\mathcal{S}_1$ and $\mathcal{S}_3$ because the sum of maximum coverage weights for all class pairs obtained using attributes from subset $\mathcal{S}_2$ is largest.

where $N_\mathcal{S}$ is the number of attributes in $\mathcal{S}$. Note that the weighted maximum coverage problem is reduced to the well studied set-cover problem when all the coverage weights are equal to be ones.

**Proposition 3.2.** *The weighted maximum coverage function $\mathcal{Q} : 2^M \to R$ is a monotonically increasing submodular function under the proposed set representation.*

For the weighted maximum coverage term, monotonicity is obvious because the addition of any attribute will increase the number of covered elements in $\mathcal{U}$. Submodularity results from the observation that the coverage weights of increased covered elements will be less from adding an attribute in a later stage because some elements may be already covered by previously selected attributes. The proof is given in the supplementary section.

### 3.4  Objective Function and Optimization

Combing the entropy rate term and the weighted maximum coverage term, the overall objective function for attribute selection is formulated as follows:

$$\max \mathcal{F}(\mathcal{S}) = \max_S \mathcal{H}(\mathcal{S}) + \lambda \mathcal{Q}(\mathcal{S}) \ \text{s.t.} \ N_\mathcal{S} \leq K \tag{4}$$

where $\lambda$ controls the relative contribution between entropy rate and the weighted maximum coverage term. The objective function is submodular because linear combination of two submodular functions with nonnegative coefficients preserves submodularity [24]. Direct maximization of a submodular

---

**Algorithm 1** Submodular Attribute Selection

---

1: **Input:** $G = (V, E)$, $A$ and $\lambda$
2: **Output:** $\mathcal{S}$
3: **Initialization**: $\mathcal{S} \leftarrow \emptyset$
4: **for** $N_\mathcal{S} < K$ and $F(S \cup a) - F(S) \geq 0$ **do**
5: $\quad a_m = argmax_{\mathcal{S} \cup a_m} \mathcal{F}(\mathcal{S} \cup \{a_m\}) - \mathcal{F}(\mathcal{S})$
6: $\quad \mathcal{S} \leftarrow a_m$
7: **end for**

---

function is an NP-hard problem. However, a greedy algorithm from [24] gives a near-optimal solution with a $(1 - 1/e)$-approximation bound. The greedy algorithm starts from an empty attribute set $\mathcal{S} = \emptyset$ ; and iteratively adds one attribute that provides the largest gain for $\mathcal{F}$ at each iteration. The iteration stops when the maximum number of selected attributes is obtained or $\mathcal{F}(\mathcal{S})$ decreases. Algorithm 1 presents the pseudo code of our algorithm. A naive implementation of this algorithm has the complexity of $O(|M|^2)$, because it needs to loop $O(|M|)$ times to add a new attribute and scan through the whole attribute list in each loop. By exploiting the submodularity of the objective function, we use the lazy greedy approach presented in [16] to speed up the optimization process.

### 3.5  Human-labeled Attribute and Data-driven Attribute Extraction

Action videos can be characterized by a collection of human-labeled attributes [20]. For example, the action "long-jump" in Olympic Sports Dataset [25] is associated with either the motion attributes (*jump forward, motion in the air*), or with the scene attributes (e.g., *outdoor, track*). Given an action

video $x$, an attribute classifier $f_a : x \rightarrow \{0, 1\}$ predicts the confidence score of the presence of attribute $a$ in the video. This classifier $f_a$ is learned using the training samples of all action classes which have this attribute as positive and the rest as negative. Given a set of attribute classifiers $S = \{f_{a_i}(x)\}_{i=1}^m$, an action video $x \in R^d$ is mapped to the semantic space $\mathcal{O}$: $h : R^d \rightarrow \mathcal{O} = [0, 1]^m$ where $h(x) = (h_1(x), ..., h_m(x))^T$ is a $m$-dimensional attribute score vector.

Previous works [21, 20] on data-driven attribute discovery used $k$-means or information theoretic clustering algorithms to obtain the clusters as the learned attributes. In this paper, we propose to discover a large initial set of data-driven attributes using a dictionary learning method. Specifically, assume that we have a set of N videos in a $n$-dimensional feature space $X = [x_1, ..., x_N], x_i \in R^n$, then a data-driven dictionary is learned by solving the following problem:

$$\arg \min_{D, Z} ||X - DZ||_2^2 \ s.t. \ \forall i, \ ||z_i||_0 \leq T \tag{5}$$

where $D = [d_1 ... d_K], d_i \in R^n$ is the learned attribute dictionary of size $K$, $Z = [z_i ... z_N], z_i \in R^K$ are the sparse codes of $X$, and $T$ specifies the sparsity that each video has fewer than $T$ items in its decomposition. Compared to $k$-means clustering, this dictionary-based learning scheme avoids the hard assignment of cluster centers to data points. Meanwhile, it doesn't require the estimation of the probability density function of clusters in information theoretic clustering. Note that our attribute selection framework is very general and different initial attribute extraction methods can be used here.

## 4 Experiments

In this section, we validate our method for action recognition on two public datasets: Sports dataset [25] and UCF101 [20] dataset. Specifically, we consider three sets of attributes: human-labeled attribute set (**HLA set**), data-driven attribute set (**DDA set**) and the set mixing both types of attributes (**Mixed set**). To demonstrate the effectiveness of our selection framework, we compare the result using the selected subset with the result based on the initial set.

We also compare our method with other two submodular approaches based on the *facility location* function (FL) and *saturated coverage* function (SC) respectively in [23]. These objective functions are defined as follows: $\mathcal{F}_{fa}(\mathcal{S}) = \sum_{i \in V} \max_{j \in S} w_{i,j}$, $\mathcal{F}_{sa}(\mathcal{S}) = \sum_{i \in V} \min\{C_i(\mathcal{S}), \alpha C_i(\mathcal{V})\}$ where $w_{i,j}$ is a similarity between attribute $i$ and $j$, $C_i(\mathcal{S}) = \sum_{j \in \mathcal{S}} w_{i,j}$ measures the degree that attribute $i$ is "covered" by $\mathcal{S}$ and $\alpha$ is a hyperparameter that determines a global saturation threshold. For the two approaches compared against, we consider an undirected $k$-nearest neighbor graph and use a Gaussian kernel to compute pairwise similarities $w_{i,j} = \exp(-\beta d_{i,j}^2)$ where $d_{i,j}$ is the distance between attribute $i$ and $j$, $\beta = (2\langle d_{i,j}^2 \rangle)^{-1}$ and $\langle \cdot \rangle$ denotes expectation over all pairwise distances.

Finally, we compare the performance of attribute-based representation with several state-of-the-art approaches on the two datasets.

### 4.1 Olympic Sports Dataset

The Olympic Sports dataset contains 783 YouTube video clips of 16 sports activities. We followed the protocol in [20] to extract STIP features [4]. Each action video is finally represented by a 2000-dimensional histogram. We use 40 human-labeled attributes provided by [20]. Three attribute-based representations are constructed as follows: (1) **HLA set**: For each human-labeled attribute, we train a binary SVM with a histogram intersection kernel. We concatenate confidence scores from all these attribute classifiers into a 40-dimensional vector to represent this video. (2) **DDA set**: For data-driven attributes, we learn a dictionary of size 457 from all video features using KSVD [1] and each video is represented by a 457-dimensional sparse coefficient vector. (3) **Mixed set**: This attribute set is obtained by combining **HLA set** and **DDA set**.

We compare the performance of features based on selected attributes with those based on the initial attribute set. For all the different attribute-based features, we use an SVM with Gaussian kernel for classification. Table 1 shows classification accuracies of different attribute-based representations. Compared with the initial attribute set, the selected attributes have greatly improved the classification accuracy, which demonstrates the effectiveness of our method for selecting a subset of discriminative attributes. Moreover, features based on the **Mixed** set outperform features based on either **HLA** set or **DDA** set. This shows that data-driven attributes are complementary to human-labeled attributes and together they offer a better description of actions. Table 2 shows the per-category average precision (AP) and mean AP of different approaches. It can be seen that our method achieves

| dataset | HLA | | DDA | | Mixed | |
|---|---|---|---|---|---|---|
| | **All** | **Subset** | **All** | **Subset** | **All** | **Subset** |
| Olympic | 61.8 | **64.1** | 49.0 | **53.8** | 63.1 | **66.7** |
| UCF101 | 81.7 | **83.4** | 79.0 | **81.6** | 82.3 | **85.2** |

Table 1: **Recognition results of different attribute-based representations**. "All" denotes the original attribute sets and "Subset" denote the selected subsets.

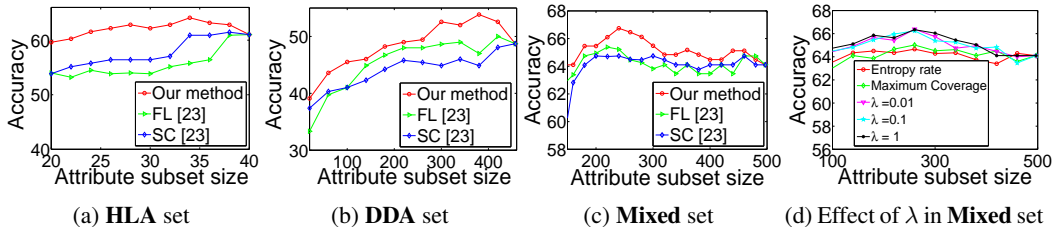

(a) **HLA** set     (b) **DDA** set     (c) **Mixed** set     (d) Effect of $\lambda$ in **Mixed** set

Figure 4: **Recognition results by different submodular methods on the Olympic Sports dataset.**

| Activity | [15] | [25] | [32] | [20] | [17] | **HLA** | **DDA** | **Mixed** |
|---|---|---|---|---|---|---|---|---|
| high-jump | 52.4 | 68.9 | 18.4 | **93.2** | 82.2 | 80.4 | 66.4 | 83.1 |
| long-jump | 66.8 | 74.8 | 81.8 | 82.6 | 92.5 | 88.8 | 85.3 | **93.9** |
| triple-jump | 36.1 | 52.3 | 16.1 | 48.3 | 52.1 | 61.4 | 60.7 | **73.6** |
| pole-vault | 47.8 | 82.0 | **84.9** | 74.4 | 79.4 | 55.1 | 45.5 | 56.8 |
| gym. vault | 88.6 | 86.1 | 85.7 | 86.7 | 83.4 | 98.2 | 84.2 | **98.4** |
| short-put | 56.2 | 62.1 | 43.3 | **76.2** | 70.3 | 63.7 | 39.5 | 72.2 |
| snatch | 41.8 | 69.2 | **88.6** | 71.6 | 72.7 | 74.5 | 34.2 | 79.8 |
| clean-jerk | 83.2 | 84.1 | 78.2 | 79.4 | 85.1 | 73.8 | 57.9 | **82.6** |
| javelin throw | 61.1 | 74.6 | 79.5 | 62.1 | **87.5** | 36.0 | 26.4 | 36.5 |
| hammer throw | 65.1 | 77.5 | 70.5 | 65.5 | 74.0 | 76.9 | 77.2 | **80.4** |
| discuss throw | 37.4 | 58.5 | 48.9 | **68.9** | 57.0 | 53.9 | 45.6 | 56.0 |
| diving-plat. | 91.5 | 87.2 | 93.7 | 77.5 | 86.0 | 94.8 | 55.3 | **99.2** |
| diving-sp. bd. | 80.7 | 77.2 | 79.3 | 65.2 | 78.3 | 79.7 | 59.7 | **90.4** |
| bask. layup | 75.8 | 77.9 | 85.5 | 66.7 | 78.1 | 88.7 | 89.7 | **90.7** |
| bowling | 66.7 | **72.7** | 64.3 | 72.0 | 52.5 | 43.0 | 55.3 | 55.4 |
| tennis-serve | 39.6 | 49.1 | 49.6 | 55.2 | 38.7 | 78.8 | 35.3 | **83.7** |
| **mean-AP** | 62.0 | 72.1 | 66.8 | 71.6 | 73.2 | 72.1 | 57.2 | **77.0** |

Table 2: **Average precisions for activity recognition on the Olympic Sporst dataset.**

the best performance. This illustrates the benefits of selecting discriminative attributes and removing noisy and redundant attributes. Note that our method outperforms the method that is most similar to ours [20] which uses complex latent SVMs to combine low-level features, human-labeled attributes and data-driven attributes. Moreover, compared with other dynamic classifiers [25, 17] which account for the dynamics of bag-of-features or action attributes, our method still obtains comparable results. This is because the provided human-labeled attributes are very noisy and they can greatly affect the training of latent SVM and representation of the attribute dynamics.

Figures 4a 4b 4c show classification accuracies of attribute subsets selected by different submodular selection methods. It can be seen that our method outperforms the other two submodular selection methods for the three different attribute sets. This is because our method prefers attributes with large and similar discrimination capability for differentiating pairwise classes, while the other two methods prefer attributes with large similarity to other attributes (i.e. representative), without explicitly considering the discrimination capabilities of selected attributes. Figure 4d shows the performance curves for a range of $\lambda$. We observe that the combination of entropy rate term and maximum coverage term obtains a higher classification accuracy than when only one of them is used. In addition, our approach is insensitive to the selection of $\lambda$. Hence we use $\lambda = 0.1$ throughout the experiments.

## 4.2 UCF101 Dataset

UCF101 dataset contains over 10,000 video clips from 101 different human action categories. We compute the improved version of dense trajectories in [34] and extract three types of descriptors: histogram of oriented gradients (HOG), histogram of optical flow (HOF) and motion boundary his-

| splits | [34] | [36] | [37] | [11] | [29] | **HLA** | **DDA** | **Mixed** |
|---|---|---|---|---|---|---|---|---|
| 1 | 83.03 | 83.11 | 79.41 | 65.22 | 63.41 | 82.45 | 80.35 | **84.19** |
| 2 | 84.22 | 84.60 | 81.25 | 65.39 | 65.37 | 83.27 | 82.16 | **85.51** |
| 3 | 84.80 | 84.23 | 82.03 | 67.24 | 64.12 | 84.60 | 82.42 | **86.30** |
| Avg | 84.02 | 83.98 | 80.90 | 65.95 | 64.30 | 83.44 | 81.64 | **85.24** |

Table 3: **Recognition results of different approaches on UCF101 dataset.**

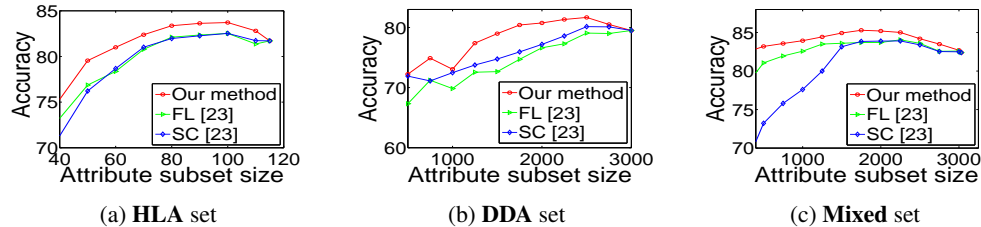

(a) **HLA** set         (b) **DDA** set         (c) **Mixed** set

Figure 5: **Recognition results by different submodular methods on UCF101 dataset.**

togram (MBH). We use Fisher vector encoding [26] and obtain 101,376-dimensional histogram to represent each action video. Three different attribute sets and corresponding attribute-based representations are constructed as follows: (1) **HLA set**: Due to the high dimensionality of features and large number of samples, the linear SVM is trained for the detection of each human-labeled attribute. We concatenate confidence scores from all these attribute classifiers into a 115-dimensional vector to represent a video. (2) **DDA set**: For data-driven attributes, we first apply PCA to reduce the dimension of histogram descriptors to be 3300 and then learn a dictionary of size 3030. The features based on data-driven attributes are 3030-dimensional sparse coefficient vectors. (3) **Mixed set**: **HLA set** plus **DDA set**.

Following the training and testing dataset partitions proposed in [30], we train a linear SVM and report classification accuracies of different attribute-based representations in Table 1. The selected attribute subset outperforms the initial attribute set again which demonstrates the effectiveness of our proposed attribute selection method. Figure 5 shows the results of attribute subsets selected by different submodular selection methods. Note that this dataset is highly challenging because the training and test videos of the same action have different backgrounds and actors. You can see that our method still substantially outperforms the other two submodular methods. This is because some redundant attributes dominated the selection process and the attributes selected by comparing approaches had very unbalanced discrimination capability for different classes. However, the attributes selected by our method have strong and similar discrimination capability for each class. Table 3 presents the classification accuracies of several state-of-the-art approaches on this dataset. Our method achieves comparable results to the best result 85.9% from [34] which uses complex spatio-temporal pyramids to embed structure information in features. Note that our method also outperforms other methods which make use of complicated and advanced feature extraction and encoding techniques.

## 5 Conclusion

We exploited human-labeled attributes and data-driven attributes for improving the performance of action recognition algorithms. We first presented three attribute selection criteria for the selection of discriminative and compact attributes. Then we formulated the selection procedure as one of optimizing a submodular function based on the entropy rate of a random walk and weighted maximum coverage function. Our selected attributes not only have strong and similar discrimination capability for all pairwise classes, but also maximize the sum of largest discrimination capability that each pairwise classes can obtain from the selected attributes. Experimental results on two challenging dataset show that the proposed method significantly outperforms many state-of-the art approaches.

## 6 Acknowledgements

The identification of any commercial product or trade name does not imply endorsement or recommendation by NIST. This research was partially supported by a MURI from the Office of Naval research under the Grant 1141221258513.

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
