[Supplementary Material]

# Submodular Attribute Selection for Action Recognition in Video: Supplementary Material

**Jinging Zheng**
UMIACS, University of Maryland
College Park, MD, USA
zjngjng@umiacs.umd.edu

**Zhuolin Jiang**
Noah's Ark Lab
Huawei Technologies
zhuolin.jiang@huawei.com

**Rama Chellappa**
UMIACS, University of Maryland
College Park, MD, USA
rama@umiacs.umd.edu

**P. Jonathon Phillips**
National Institute of Standards and Technology
Gaithersburg, MD, USA
jonathon.phillips@nist.gov

## 1 Proof of Submodularity of Entropy Rate

Recall our definition of $\mathcal{H}(\mathcal{S})$:

$$\mathcal{H}(\mathcal{S}) = -\sum_i u_i \sum_j p_{i,j}(\mathcal{S}) log(p_{i,j}(\mathcal{S})) \tag{1}$$

where $u_i$ is the stationary probability of $v_i$ in the stationary distribution and $p_{i,j}(\mathcal{S})$ is the transition probability from $v_i$ to $v_j$ with respect to $\mathcal{S}$. T

*Proof.* We prove the submodularity by showing

$$\mathcal{H}(\mathcal{S} \cup \{a_1\}) - \mathcal{H}(\mathcal{S}) \geq \mathcal{H}(\mathcal{S} \cup \{a_1, a_2\}) - \mathcal{H}(\mathcal{S} \cup \{a_2\}). \tag{2}$$

It is known that the transition probability with respect to $\mathcal{S}$ is given as follows:

$$p_{i,j}(\mathcal{S}) = \begin{cases} \frac{w_{i,j}}{w_i} = \frac{\sum_{d\in\mathcal{S}} A_{d,l}}{w_i} & \text{if } i \neq j \\ \frac{w_{i,i}}{w_i} = \frac{\sum_{d\in\mathcal{P}\setminus\mathcal{S}} A_{d,l}}{w_i} & \text{if } i = j \end{cases} \tag{3}$$

where $w_i = \sum_{m:e_{i,m}\in E} w_{i,m}$ is the sum of incident weights of the vertex $v_i$ and $w_{i,i} = w_i - \sum_{j\neq i} w_{i,j}$, $l$ is the index of the combination of pairwise classes $(i,j)$ in $\mathcal{U}$. Without loss of generality, we assume that after the addition of attribute $a_n$ into $\mathcal{S}$, the transition probability becomes

$$p_{i,j}(\mathcal{S} \cup \{a_1\}) = \begin{cases} \frac{w_{i,j}}{w_i} + \frac{A_{n,l}}{w_i} & \text{if } i \neq j \\ \frac{w_{i,i}}{w_i} - \frac{\sum_{j\neq i} A_{n,l}}{w_i} & \text{if } i = j. \end{cases} \tag{4}$$

For simplicity of notation, we let $p_{i,j}(\mathcal{S}) = p_{i,j}$ and $p_{i,j}(\mathcal{S} \cup \{a_n\}) = p_{i,j} + \Delta_{i,j}^n, n = 1, 2$, where $\Delta_{i,j}^n$ is symmetric, i.e. $\Delta_{i,j}^n = \Delta_{j,i}^n$. We note that $\Delta_{i,j\neq i}^n \geq 0$ and $\Delta_{i,i}^n = -\sum_{j\neq i}\Delta_{i,j}^n \leq 0$. $\Delta_{i,j}^n = 0$ means that the addition of $a_n$ doesn't increase the edge weight $e_{i,j}$ while $\Delta_{i,j}^n > 0$ means that the addition of $a_n$ increase $w_{i,j}$. Similarly, we let $p_{i,j}(\mathcal{S} \cup \{a_1, a_2\}) = p_{i,j} + \Delta_{i,j}^1 + \Delta_{i,j}^2$.

$$\mathcal{H}(\mathcal{S} \cup \{a_1\}) - \mathcal{H}(\mathcal{S}) \tag{5}$$

$$= -\sum_i u_i \sum_j (p_{i,j} + \Delta_{i,j}) \log((p_{i,j} + \Delta_{i,j}) + \sum_i u_i \sum_j p_{i,j} \log p_{i,j} \tag{6}$$

$$= -\sum_i \sum_j \frac{w_i(p_{i,j} + \Delta_{i,j})}{w_0} \log(p_{i,j} + \Delta_{i,j}) - \sum_i \sum_j \frac{w_i(p_{i,j} + \Delta_{i,j})}{w_0} \log \frac{w_i}{w_0} \tag{7}$$

$$+ \sum_i \sum_j \frac{w_i p_{i,j}}{w_0} \log \frac{w_i}{w_0} + \sum_i \sum_j \frac{w_i p_{i,j}}{w_0} \log p_{i,j} \tag{8}$$

$$= -\sum_i \sum_j \frac{w_i(p_{i,j} + \Delta_{i,j})}{w_0} \log \frac{w_i(p_{i,j} + \Delta_{i,j})}{w_0} + \sum_i \sum_j \frac{w_i p_{i,j}}{w_0} \log \frac{w_i p_{i,j}}{w_0} \tag{9}$$

$$= -\sum_i \sum_j \frac{w_i p_{i,j}}{w_0} \log \frac{w_i(p_{i,j} + \Delta_{i,j})}{w_0} + \sum_i \sum_j \frac{w_i p_{i,j}}{w_0} \log \frac{w_i p_{i,j}}{w_0} - \sum_i \sum_j \frac{w_i \Delta_{i,j}}{w_0} \log \frac{w_i(p_{i,j} + \Delta_{i,j})}{w_0} \tag{10}$$

Now we prove the first two terms and the las term are larger than zeros respectively.

$$-\sum_i \sum_j \frac{w_i p_{i,j}}{w_0} \log \frac{w_i(p_{i,j} + \Delta_{i,j})}{w_0} + \sum_i \sum_j \frac{w_i p_{i,j}}{w_0} \log \frac{w_i p_{i,j}}{w_0} \tag{11}$$

$$= \sum_i \sum_j \frac{w_i p_{i,j}}{w_0} \log \frac{\frac{w_i p_{i,j}}{w_0}}{\frac{w_i(p_{i,j} + \Delta_{i,j})}{w_0}} \tag{12}$$

$$\geq \sum_i \sum_j \frac{w_i p_{i,j}}{w_0} \log \frac{\sum_i \sum_j \frac{w_i p_{i,j}}{w_0}}{\sum_i \sum_j \frac{w_i(p_{i,j} + \Delta_{i,j})}{w_0}} \tag{13}$$

$$= \sum_i \sum_j \frac{w_i p_{i,j}}{w_0} \log 1 = 0 \tag{14}$$

by the definition of transition probability $\sum_j (p_{i,j} + \Delta_{i,j}) = \sum_j p_{i,j} = 1$ and the *Log-sum inequality* stated as follows.

**Proposition 1.1.** *(Log-sum inequality) For non-negative numbers $a_1, a_2, ..., a_n$ and $b_1, b_2, ..., b_n$*

$$\sum_{i=1}^n a_i \log \frac{a_i}{b_i} \geq (\sum_{i=1}^n a_i) \log \frac{\sum_{i=1}^n a_i}{sum_{n=1}^n b_i} \tag{15}$$

*with equality if and only if $\frac{a_i}{b_i}$ = constant.*

$$-\sum_i \sum_j \frac{w_i \Delta_{i,j}}{w_0} \log \frac{w_i(p_{i,j} + \Delta_{i,j})}{w_0} \tag{16}$$

$$= -\sum_i \sum_{j \neq i} \frac{w_i \Delta_{i,j}}{w_0} \log \frac{w_i(p_{i,j} + \Delta_{i,j})}{w_0} - \sum_i \frac{w_i \Delta_{i,i}}{w_0} \log \frac{w_i(p_{i,i} + \Delta_{i,i})}{w_0} \tag{17}$$

$$= -\sum_i \sum_{j \neq i} \frac{w_i \Delta_{i,j}}{w_0} \log \frac{w_i(p_{i,j} + \Delta_{i,j})}{w_0} + \sum_i \sum_{j \neq i} \frac{w_i \Delta_{i,j}}{w_0} \log \frac{w_i(p_{i,i} + \Delta_{i,i})}{w_0} \tag{18}$$

$$= \sum_i \sum_{j \neq i} \frac{w_i \Delta_{i,j}}{w_0} \log \frac{p_{i,i} + \Delta_{i,i}}{p_{i,j} + \Delta_{i,j}} \tag{19}$$

## 1.1 Submodularity

*Proof.* We prove the submodularity by showing

$$\mathcal{H}(\mathcal{S} \cup \{a_1\}) - \mathcal{H}(\mathcal{S}) \geq \mathcal{H}(\mathcal{S} \cup \{a_1, a_2\}) - \mathcal{H}(\mathcal{S} \cup \{a_2\}). \tag{20}$$

Similarly, for simplicity of notation, we let $p_{i,j}(\mathcal{S} \cup \{a_1\}) = p_{i,j} + \Delta_{i,j}^1$ and $p_{i,j}(\mathcal{S} \cup \{a_1, a_2\}) = p_{i,j} + \Delta_{i,j}^1 + \Delta_{i,j}^2$.

$$\mathcal{H}(\mathcal{S} \cup \{a_1\}) - \mathcal{H}(\mathcal{S}) - \mathcal{H}(\mathcal{S} \cup \{a_1, a_2\}) + \mathcal{H}(\mathcal{S} \cup \{a_2\}) \tag{21}$$

$$= -\sum_i \sum_j \frac{w_i(p_{i,j} + \Delta_{i,j}^1)}{w_0} \log \frac{w_i(p_{i,j} + \Delta_{i,j}^1)}{w_0} + \sum_i \sum_j \frac{w_i p_{i,j}}{w_0} \log \frac{w_i p_{i,j}}{w_0} \tag{22}$$

$$+ \sum_i \sum_j \frac{w_i(p_{i,j} + \Delta_{i,j}^1 + \Delta_{i,j}^2)}{w_0} \log \frac{w_i(p_{i,j} + \Delta_{i,j}^1 + \Delta_{i,j}^2)}{w_0} - \sum_i \sum_j \frac{w_i(p_{i,j} + \Delta_{i,j}^2)}{w_0} \log \frac{w_i(p_{i,j} + \Delta_{i,j}^2)}{w_0} \tag{23}$$

$$= -\sum_i \sum_j \frac{w_i(p_{i,j} + \Delta_{i,j}^1)}{w_0} \log \frac{w_i(p_{i,j} + \Delta_{i,j}^1)}{w_0} + \sum_i \sum_j \frac{w_i(p_{i,j} + \Delta_{i,j}^1)}{w_0} \log \frac{w_i(p_{i,j} + \Delta_{i,j}^1 + \Delta_{i,j}^2)}{w_0} \tag{24}$$

$$+ \sum_i \sum_j \frac{w_i \Delta_{i,j}^2}{w_0} \log \frac{w_i(p_{i,j} + \Delta_{i,j}^1 + \Delta_{i,j}^2)}{w_0} - \sum_i \sum_j \frac{w_i(p_{i,j} + \Delta_{i,j}^2)}{w_0} \log \frac{w_i(p_{i,j} + \Delta_{i,j}^2)}{w_0} \tag{25}$$

$$+ \sum_i \sum_j \frac{w_i p_{i,j}}{w_0} \log \frac{w_i p_{i,j}}{w_0} \tag{26}$$

$$= -\sum_i \sum_j \frac{w_i(p_{i,j} + \Delta_{i,j}^1)}{w_0} \log \frac{w_i(p_{i,j} + \Delta_{i,j}^1)}{w_0} + \sum_i \sum_j \frac{w_i(p_{i,j} + \Delta_{i,j}^1)}{w_0} \log \frac{w_i(p_{i,j} + \Delta_{i,j}^1 + \Delta_{i,j}^2)}{w_0} \tag{27}$$

$$+ \sum_i \sum_j \frac{w_i(p_{i,j} + \Delta_{i,j}^2)}{w_0} \log \frac{w_i(p_{i,j} + \Delta_{i,j}^1 + \Delta_{i,j}^2)}{w_0} - \sum_i \sum_j \frac{w_i(p_{i,j} + \Delta_{i,j}^2)}{w_0} \log \frac{w_i(p_{i,j} + \Delta_{i,j}^2)}{w_0} \tag{28}$$

$$+ \sum_i \sum_j \frac{w_i p_{i,j}}{w_0} \log \frac{w_i p_{i,j}}{w_0} - \sum_i \sum_j \frac{w_i p_{i,j}}{w_0} \log \frac{w_i(p_{i,j} + \Delta_{i,j}^1 + \Delta_{i,j}^2)}{w_0} \tag{29}$$

$$= \sum_i \sum_j \frac{w_i(p_{i,j} + \Delta_{i,j}^1)}{w_0} \log \frac{\frac{w_i(p_{i,j} + \Delta_{i,j}^1 + \Delta_{i,j}^2)}{w_0}}{\frac{w_i(p_{i,j} + \Delta_{i,j}^1)}{w_0}} \tag{30}$$

$$+ \sum_i \sum_j \frac{w_i(p_{i,j} + \Delta_{i,j}^2)}{w_0} \log \frac{\frac{w_i(p_{i,j} + \Delta_{i,j}^1 + \Delta_{i,j}^2)}{w_0}}{\frac{w_i(p_{i,j} + \Delta_{i,j}^2)}{w_0}} \tag{31}$$

$$+ \sum_i \sum_j \frac{w_i p_{i,j}}{w_0} \log \frac{\frac{w_i p_{i,j}}{w_0}}{\frac{w_i(p_{i,j} + \Delta_{i,j}^1 + \Delta_{i,j}^2)}{w_0}} \tag{32}$$

$$\geq \sum_i \sum_j \frac{w_i(p_{i,j} + \Delta_{i,j}^1)}{w_0} \log \frac{\sum_i \sum_j \frac{w_i(p_{i,j} + \Delta_{i,j}^1 + \Delta_{i,j}^2)}{w_0}}{\sum_i \sum_j \frac{w_i(p_{i,j} + \Delta_{i,j}^1)}{w_0}} \tag{33}$$

$$+ \sum_i \sum_j \frac{w_i(p_{i,j} + \Delta_{i,j}^2)}{w_0} \log \frac{\sum_i \sum_j \frac{w_i(p_{i,j} + \Delta_{i,j}^1 + \Delta_{i,j}^2)}{w_0}}{\sum_i \sum_j \frac{w_i(p_{i,j} + \Delta_{i,j}^2)}{w_0}} \tag{34}$$

$$+ \sum_i \sum_j \frac{w_i p_{i,j}}{w_0} \log \frac{\sum_i \sum_j \frac{w_i p_{i,j}}{w_0}}{\frac{\sum_i \sum_j w_i(p_{i,j} + \Delta_{i,j}^1 + \Delta_{i,j}^2)}{w_0}} \tag{35}$$

$$= \sum_i \sum_j \frac{w_i(p_{i,j} + \Delta_{i,j}^1)}{w_0} \log 1 + \sum_i \sum_j \frac{w_i(p_{i,j} + \Delta_{i,j}^2)}{w_0} \log 1 + \sum_i \sum_j \frac{w_i p_{i,j}}{w_0} \log 1 \tag{36}$$

$$= 0. \tag{37}$$

by the definition of the transition probability

$$\sum_j p_{i,j} = \sum_j (p_{i,j} + \Delta_{i,j}^1) = \sum_j (p_{i,j} + \Delta_{i,j}^1 + \Delta_{i,j}^2) = 1 \tag{38}$$

## 2  Proof of Proposition 2

The proof contains two parts. The first part proves $\mathcal{Q}(\mathcal{S})$ is monotonically increasing. In the second part, we show that $\mathcal{Q}(\mathcal{S})$ is submodular.

### 2.1  Proof of the monotonically increasing property

*Proof.* Let $\mathcal{S}$ be a subset of attributes and $a_1 \in \mathcal{P}$ be any attribute. We prove the monotonically increasing property

$$\mathcal{Q}(\mathcal{S} \cup \{a_1\}) - \mathcal{Q}(\mathcal{S}) \geq 0. \tag{39}$$

$$\mathcal{Q}(\mathcal{S} \cup \{a_1\}) - \mathcal{Q}(\mathcal{S}) = \sum_{u_l \in \mathcal{U}} \max_{d \in \mathcal{S} \cup \{a_1\}} A_{d,l} - \sum_{u_l \in \mathcal{U}} \max_{d \in \mathcal{S}} A_{d,l} \tag{40}$$

$$= \sum_{u_l \in \mathcal{U}} [\max(\max_{d \in \mathcal{S}} A_{d,l}, A_{1,l}) - \max_{d \in \mathcal{S}} A_{d,l}] \geq 0 \tag{41}$$

### 2.2  Proof of the submodularity

*Proof.* We prove the submodularity by showing

$$\mathcal{Q}(\mathcal{S} \cup \{a_1\}) - \mathcal{Q}(\mathcal{S}) \geq \mathcal{Q}(\mathcal{S} \cup \{a_1, a_2\}) - \mathcal{Q}(\mathcal{S} \cup \{a_2\}). \tag{42}$$

.

$$\mathcal{Q}(\mathcal{S} \cup \{a_1\}) - \mathcal{Q}(\mathcal{S}) \geq \mathcal{Q}(\mathcal{S} \cup \{a_1, a_2\}) - \mathcal{Q}(\mathcal{S} \cup \{a_2\}) \tag{43}$$
$$= \sum_{u_l \in \mathcal{U}} [\max(\max_{d \in \mathcal{S}} A_{d,l}, A_{1,l}) - \max_{d \in \mathcal{S}} A_{d,l} - \max(\max_{d \in \mathcal{S}} A_{d,l}, A_{1,l}, A_{2,l}) + \max(\max_{d \in \mathcal{S}} A_{d,l}, A_{2,l})]. \tag{44}$$

Depending on which term from the three terms $\max_{d \in \mathcal{S}} A_{d,l}$, $A_{1,l}$ and $A_{2,l}$ is largest, we consider three cases and prove that

$$\mathcal{Q}_l = \max(\max_{d \in \mathcal{S}} A_{d,l}, A_{1,l}) - \max_{d \in \mathcal{S}} A_{d,l} - \max(\max_{d \in \mathcal{S}} A_{d,l}, A_{1,l}, A_{2,l}) + \max(\max_{d \in \mathcal{S}} A_{d,l}, A_{2,l}) \geq 0 \tag{45}$$

for given $u_l \in \mathcal{U}$.

**Case 1**: Assume that $\max_{d \in \mathcal{S}} A_{d,l}$ is the largest, i.e. $\max_{d \in \mathcal{S}} A_{d,l} \geq A_{1,l}, \max_{d \in \mathcal{S}} A_{d,l} \geq A_{2,l}$, then

$$Q_l = \max_{d \in \mathcal{S}} A_{d,l} - \max_{d \in \mathcal{S}} A_{d,l} - \max_{d \in \mathcal{S}} A_{d,l} + \max_{d \in \mathcal{S}} A_{d,l} = 0. \tag{46}$$

**Case 2**: Assume that $A_{1,l}$ is the largest, i.e. $A_{1,l} \geq \max_{d \in \mathcal{S}} A_{d,l}, A_{1,l} \geq \max_{d \in \mathcal{S}}$, then

$$Q_l = A_{1,l} - \max_{d \in \mathcal{S}} A_{d,l} - A_{1,l} + \max_{d \in \mathcal{S}} A_{d,l} \tag{47}$$

$$= \max(\max_{d \in \mathcal{S}} A_{d,l}, A_{2,l}) - \max_{d \in \mathcal{S}} A_{d,l} \geq 0. \tag{48}$$

**Case 3**: Assume that $A_{2,l}$ is the largest, i.e. $A_{2,l} \geq \max_{d \in \mathcal{S}} A_{d,l}, A_{2,l} \geq \max_{d \in \mathcal{S}}$, then

$$Q_l = \max(\max_{d \in \mathcal{S}} A_{d,l}, A_{1,l}) - \max_{d \in \mathcal{S}} A_{d,l} - A_{2,l} + A_{2,l} \tag{49}$$

$$= \max(\max_{d \in \mathcal{S}} A_{d,l}, A_{1,l}) - \max_{d \in \mathcal{S}} A_{d,l} \geq 0. \tag{50}$$