[Reviews · NeurIPS 2014]

Submitted by Assigned_Reviewer_19

This paper presents an algorithm for attribute
(feature) selection, applied to activity recognition. The
approach defines 3 criteria for choosing attributes for
recognition: high discrimination between pairwise classes, similarity
in the amount of discrimination over all pairs of classes, and
coverage of all pairs of classes. The first two of these
criteria are formulated using a random walk approach, the latter
using a set-cover approach. Finally, a greedy optimization
strategy is used to choose an attribute subset, starting from
the empty set.

This paper presents interesting research. The main contribution of
the paper is the algorithm for feature selection. This
algorithm seems novel, and is an interesting combination of
random walk, set-cover, and greedy optimization. The
experimental results are also good -- the method shows
improvements over existing methods and baselines including no
feature selection and alternative strategies. There are some
missing details in the experiments but the results seem solid.

My main concern about the paper is the motivation/intuition for
the approach -- the pieces seem chosen to try to use submodular
optimization. The discussion on lines 133-145 describes 3
criteria. I wasn't clear on the motivation for criterion 2 --
why can't some attributes be better for certain classes than
others? Can't a final classifier choose combinations? It seems
that this second criterion complicates the optimization, and
necessitates the use of the proposed techniques.

The resulting submodular optimization amounts to a
greedy approach. It seems one could do similar greedy
optimization for other fomulations, for example ones with
criterion 2. It would be interesting to know whether this
criterion is important empirically.

Overall, this concern is not fundamental; I think the paper is
very good, and should be accepted.

Other comments:

- I haven't gone over the details in supplementary material --
the intuitive explanations in the paper for these seemed
reasonable though.

- Typo "Combing"

- I didn't understand where the sparse coding (Eq. 5) is used in
the paper. The experiments refer to KSVD for the DDA set.
Sparse coding of attributes doesn't seem to be a key component
of the paper, but if this is used it should be compared to
k-means or alternatives. (Though there isn't space for this in
a conference version.)

- How are hyperparamters set in the experiments? Are they tuned
by looking at the test (i.e. cross-validation) accuracies?

- The experimental results seem better than previous work.
Where are the numbers in Table 2 from? [20] is shown with a
per-activity AP list, and mAP of 71.6. In [20], Fig. 10(b)
shows mAP of 74.38%. This would seem to be better than what is
in this paper.

- The related work section is thorough, though the work of Fu et
al., who also do similar attribute learning could be added:

Learning multi-modal latent attributes
Y. Fu, T. Hospedales, T. Xiang and S. Gong
IEEE Transactions on Pattern Analysis and Machine Intelligence (PAMI 2013)

Summary: Novel algorithm for attribute or feature selection.
The motivation for part of the approach is not entirely clear,
but the method is novel and seems effective empirically.

Submitted by Assigned_Reviewer_31

This paper addresses the problem of action recognition in video. The authors encode actions based on two types of attributes that describe the actions as high-level concepts: 1) Human Labeled Attributes (HLA) and 2) Data Driven Attributes (DDA). The authors propose a method to select a subset of the attributes that are discriminative and compact by combining submodular objectives that promote discrimination through the entropy rate-based attribute selection and coverage through the weighted maximum coverage-based attribute selection. The authors present experiments on the Olympic Sports and UCF101 datasets, two difficult datasets for action recognition and compare to a set of baselines and state-of-the-art approaches.

The paper is mostly well written, except a few sections. The problem is of significance as it is a standard problem in computer vision, and the experiments are performed on relevant and difficult datasets. The overall formulation of the attribute selection problem as a weighted maximum coverage problem and entropy rate maximization for a random walk is novel and interesting.

The core problem tackled in this paper seems to be the feature selection problem, as the authors try to select a subset of attribute classifiers that perform well. However, this a standard problem and there are many simple approaches for doing this, such as approaches with L1 regularization like the lasso, L1 regularized hinge loss, elastic net. I would encourage the authors to make direct comparisons to these simple approaches, instead of focusing only on other submodular selection methods.

State-of-the-art results reported in other works are significantly higher due to the use of better dense feature vectors (90.2% on Olympic dataset as reported in [34], also other numbers Jiang et al. (ECCV 12), Jain et al (CVPR 13), Gaidon et al. (BMVC 12)are close to 80%). It would be interesting to see if the proposed attribute selection method leads to an improved performance, when combined with these dense features.

Further, this work proposes a general method for feature selection, and is not specific to action classes. Hence, to truly justify the potential of this work, I would encourage the authors to run experiments in other domains in vision, which use attributes as well (refer to [11] for other datasets).

There are a few mistakes in the equations, which should be corrected if the paper is selected for publication:
- The equation for A_{d,l} in L129 is not clear. What is $u^d$ and $u_k^d$ in this equation? Are these supposed to be $\mu$. What does mean and standard deviation of a "class" refer to in L127?
- Similarly, Eq. 2, u_i should perhaps be \mu_i
- L161: Define T

Summary: The overall formulation of attribute selection through the use of submodular objective functions which encourages the selection of discriminative attributes is novel and interesting. The authors address a standard feature selection problem, but restrict the experiments only to video action datasets, and do not compare to state-of-the-art results which use better features than the ones used in the paper.

Submitted by Assigned_Reviewer_38

The proposed paper would like to use attributes as a mid-level feature representation for doing action recognition (AR). Currently, the attributes used in AR are all human assigned without accounting for discriminative ability. This paper proposes a data-driven approach to "discover", or rather select a limited set of discriminative attributes from a larger set.

QUALITY:

The formulation of the attribute selection criteria seems correct. What is missing is the motivation for keeping the relationships only pairwise and not either higher-order or one-vs-all?

The overall claims made in the introduction are very much grander than what the results can substantiate. In particular, the claims of human-labelled attributes (HLA) being "arbitrary", and have noise and or redundancy which may "lead to degradation in action recognition performance" does not give enough credit to HLAs. The attributes are typically selected their semantic meaning and plays a role once hierarchies of activities are taken into place, e.g. applying eye makeup and applying licpstick both have the same attributes as they both fall into the category of applying makeup. Furthermore, as the experimental results show in tables 2 & 3, with the exception of a few classes, the data-driven attributes (DDA) are actually much weaker than HLA, with some results being significantly worse. IT is only in the "mixed" case in which there are improvements to be found, suggesting that the DDA are at best complementary to the HLA, and alone are insufficient. As such, claims that one can find a more compact set of attributes are not substantiated.

CLARITY:

line 102 - 103: while the application area is interesting, more relevant to know *HOW* they have been used or applied in such applications, e.g. for optimization, classification, etc.

line 129: what is u_k^d and u^d?

How is lambda in equation 4 determined and what impact does it have on the results?

ORIGINALITY: the paper states that they define a "novel submodular objective function for attribute selection", but how it is novel is not clear. Furthermore, please comment on what distinguishes this submodularity optimization from the other approaches listed in the related works section and especially [22] and [23] in which the method is compared.

SIGNIFICANCE:
This paper would be of interest to those who work in action recognition and or attribute representations, both of which are sub-topics in computer vision.
Summary: An interesting approach for determining data-driven attribute representations. A good paper which needs only some refinement of writing in the introduction / motivation
Author Feedback
Author rebuttal: We sincerely thank all the reviewers for their helpful comments.

Assigned_Review_19

R19 asked about the motivation of criterion 2. As described in lines 136-137, we hope to select attributes that have similar discriminability for each pairwise classes, so that we could balance the discrimination power of the selected attributes for different classes, and avoid the situation that the selected attributes are only discriminative for some classes. We will add more explanation.

R19 asked whether some attributes can be better for certain classes and a final classifier can be used for choosing their combinations. It is true that some attributes are better for certain classes, but others are not. As explained in lines 53-65, a large set of attributes may contain many redundant or irrelevant attributes, hence if we directly train a final classifier for choosing their combinations, the high dimensional attribute feature may cause overfitting and increase the training time. Our paper presents a novel approach to efficiently select a set of discriminative and compact attributes.

R19 asked about the usage of Eq. 5. It is the objective function of KSVD algorithm [1] and we use it to generate data-driven attributes.

R19 suggested to compare sparse coding with k-means or alternatives. We will add this in the final paper.

R19 asked how the hyper-parameters are set. The hyper-parameter lambda is set to 0.1 as given in line 393, and the effect of lambda is shown in Fig.4 (d).

R19 asked the source of performance statistics in Table 2 and why the result of [20] shown in Table 2 is different from that reported in [20]. The results come from two papers by the same authors, one is [17], and the other is Li et al. [CVPR13] titled “Recognizing Activities via Bag of Words for Attribute Dynamics”. Since [20] didn’t report per-activity AP performance, both [17] and the other paper re-implemented the approach in [20] and reported mAP of 71.6 instead of 74.4. We just report the result of [20] as in [17] and the other paper. This will be clarified.

R19 suggested to add one work on attribute learning, which we will add.

Assigned_Review_31

R31 suggest that we should compare with other feature selection approaches based on lasso regularization. In fact, we compared with a lasso-based method reported in [11] using the UCF101 dataset and obtained better performance. We note that our work is different from them in two aspects. First, our method is discriminative and considers the pairwise relationship between attributes; lasso-based methods are typically good for reconstruction without explicitly modelling the relationship between attributes. Second, our method is guaranteed to be at least (1-1/e)-approximation to the optimum and can be solved by a very efficient greedy algorithm while lasso-based methods are computationally inefficient when the dimension of features is very high.

R31 asked why the results from [34], Jiang et al. (ECCV), Jain et al. (CVPR) and Gaidon et al. (BMVC) on the Olympic dataset are not cited in table 2. There are two main reasons. First, different experimental settings are used. Those methods used the fixed training and test split given by [25] for experiments, while we followed [17] to used 5-fold cross-validation. Second, different features are used. Those four methods used more sophisticated features, such as dense trajectory-based features of 2*10^6 and 128*10^6 dimensions used in [34] and Jiang et al. (ECCV) respectively. Instead, we followed [17] [20] [25] and used simple STIP features [4]. When both our approach and [34] used the same feature for UCF 101 dataset, our approach is better than [34] as shown in Table 3.

R31 stated that we didn’t compare with previous data-driven attribute discovery methods such as [20, 21]. This is because finding data-driven attributes is not a key component of our paper, different methods can be used to generate data-driven attributes as stated in lines 288-290. In fact, we compared with [20] in Table 2.

R31 ask about the equation in Line 129. First, there should be an index of $k$ in the denominator and we will add this. Second, $u^d$ and $u_k^d$ refers to $\mu^d$ and $\mu_k^d$, we will correct this typo.

R31 asked about the mean and standard deviation of a "class" in line 127. $\mu_k^d$ is the mean value calculated using the d-th dimensional feature of samples from class k, $\sigma_k^d$ is the standard deviation value calculated using the d-th dimensional feature of samples from class k. We will clarify it in the paper.

R31 suggested us to discuss and cite a paper on submodular optimization, which we will add.

Assigned_Reviewer_38

R38 stated that our claim that one can find a more compact set of attributes is not substantiated. We argue that this claim is supported by the fact that the selected subset of mixed attributes denoted by “S-Mixed” outperforms the original set of mixed attributes denoted by “I-Mixed” as shown in table 1. We didn’t report the performance of I-HLA, I-DDA and I-Mixed in table 2 and 3 because they are always outperformed by S-HLA, S-DDA and S-Mixed in our experiments.

R38 asked for more details on how sub-modular optimization is used in different applications in related work. We will add more details.

R38 asked about u_k^d and u^d in line 129. For a pair of classes i and j, u_k^d is the mean value calculated using the d-th dimensional feature of samples from class k where k = i, j. While u^d is the mean value calculated using the d-th dimensional feature from both class i and j. This will be clarified.

R38 asked about the lambda. Since our approach is insensitive to the selection of lambda as shown in Fig.4 (d), we set lambda = 0.1 throughout the experiments as stated in line 393.

R38 asked how our method is distinguished from [22] and [23]. [22] targets on clustering while [23] targets on feature selection. Both methods are unsupervised but our method is supervised.